# Metalloproteinases and Tissue Inhibitors in Generalized Myasthenia Gravis. A Preliminary Study

**DOI:** 10.3390/brainsci12111439

**Published:** 2022-10-26

**Authors:** Vincenzo Di Stefano, Chiara Tubiolo, Andrea Gagliardo, Rosalia Lo Presti, Maria Montana, Massimiliano Todisco, Antonino Lupica, Gregorio Caimi, Cristina Tassorelli, Brigida Fierro, Filippo Brighina, Giuseppe Cosentino

**Affiliations:** 1Department of Biomedicine, Neuroscience and Advanced Diagnostic (BIND), University of Palermo, 90127 Palermo, Italy; 2Department of Health Promotion, Mother and Child Care, Internal Medicine and Medical Specialties, University of Palermo, 90127 Palermo, Italy; 3Department of Brain and Behavioral Sciences, University of Pavia, 27100 Pavia, Italy; 4IRCCS Mondino Foundation, 27100 Pavia, Italy

**Keywords:** matrix metalloproteinases, anti-AChR antibodies, myasthenia gravis, neuromuscular junction

## Abstract

Introduction: Matrix metalloproteinases (MMPs) and tissue inhibitors of metalloproteinases (TIMPs) have recently been proposed as promising biomarkers in different immune-mediated disorders. We evaluated the plasma levels of MMP-9 and MMP-2 and their tissue inhibitors TIMP-1 and TIMP-2 in a patients’ cohort with generalized myasthenia gravis (MG). Methods: Plasma concentrations of MMP-9, MMP-2, TIMP-1 and TIMP-2 were evaluated in 14 patients with generalized MG and 13 age- and sex-matched healthy controls. The severity of disease was assessed by the modified Osserman classification. Results: Compared to the healthy subjects, MG patients had increased plasma concentrations of MMP-9, but reduced plasma levels of MMP-2 and TIMP-1. MG patients also showed a positive correlation between MMP-2 concentrations and disease severity. An increase in MMP-9 levels and MMP-9/TIMP-1 ratio and a decrease in MMP-2 levels and MMP-2/TIMP-2 ratio were detected in patients with generalized MG. Higher levels of MMP-2 correlated with greater disease severity. Discussion: Our preliminary findings suggest that MMPs and TIMPs could play a role in the pathogenesis of MG and might be associated with the risk of clinical deterioration.

## 1. Introduction

Matrix metalloproteinases (MMPs) are a heterogeneous group of zinc-dependent endopeptidases with different substrate affinity for several components of the extracellular matrix (ECM), cytokines, and cell motility factors [1,2]. MMPs are synthesized as proenzymes and activated by specific proteases, becoming capable of ECM degradation [1]. Specific tissue inhibitors of metalloproteinases (TIMPs) form complexes with MMPs pro-forms and mediate the inhibition of MMPs [1,2,3]. Therefore, the balance between MMPs and TIMPs regulates the ECM turnover and an increased level of active MMPs may reflect an imbalance between pro-MMPs and TIMPs [1,4]. MMPs are involved in organ development and repair after injury in skeletal muscles and nerves [2,5,6]. Among MMPs, the gelatinases MMP-9 and MMP-2 are supposed to be involved in the mechanisms of cell migration into peripheral nerves and muscles. In particular, MMP-9 has been shown to have a role in peripheral myelination [4,7]. MMP-9 and MMP-2 are inhibited by TIMP-1 and TIMP-2, respectively [4,8].

Under physiological conditions, MMP-9 is usually detectable in the serum, but not in the cerebrospinal fluid, while MMP-2 is constitutively present in several body tissues and fluids [9,10]. MMP-9 levels can increase in pathological conditions, such as multiple sclerosis and meningitis [6], and may become detectable also in the cerebrospinal fluid of patients with inflammatory diseases of the central and peripheral nervous system [10,11,12].

It has been hypothesized that MMPs could be involved in the autoimmune response since the degradation of the ECM may cause the exposition of antigens that could trigger and favor autoimmunity [13]. Accordingly, an up-regulation of gelatinases has been found in nerve biopsy specimens from patients with inflammatory and vasculitic neuropathies [4,9]. In particular, double labeling with CD3 and CD8 revealed that most of MMP-9-positive cells are T-lymphocytes and occasionally macrophages [9]. Moreover, polymorphisms of the MMP-9 gene have been associated with higher risk of developing multiple sclerosis [14,15].

Myasthenia gravis (MG) is an autoimmune disorder with fluctuating weakness in the skeletal muscles, caused by specific autoantibodies targeting the acetylcholine receptor (AChR), or less frequently against the muscle specific kinase (MuSK) or the low-density lipoprotein receptor-related protein 4 (LRP4) at the neuromuscular junction [16,17]. Many studies have showed that anti-AChR antibodies serum levels do not correlate with the risk of occurrence of myasthenic crises or with the response to treatment [17,18]. Hence, there are no biomarkers for disease activity in MG to date.

A pathogenetic role of MMPs has been suggested for inflammatory neuromuscular diseases, but the evidence is still very limited [4,6,19]. Elevated levels of MMP-2, MMP-9 and their inhibitors have been observed in patients affected by chronic immune-mediated neuropathies and myopathies, and these levels were influenced by the immunoglobulin therapy [4]. Helgeland et al. examined the levels of MMP-2, MMP-3 and MMP-9 in patients with generalized and ocular MG, reporting elevated MMP-2 and MMP-9 levels [19]. However, TIMPs levels and correlations with clinical findings were not assessed [17].

Therefore, there is still little evidence in the literature that MMPs might have a role in the induction or maintenance of the autoimmune processes underlying MG [4,19].

In this preliminary study, we evaluated the plasma levels of MMP-2, MMP-9 and their inhibitors TIMP-1 and TIMP-2 in a patients’ group with generalized MG compared to healthy controls. Of note, we aimed to investigate whether these biological parameters could represent useful biomarkers for MG.

## 2. Methods

### 2.1. Patient Population

We included patients with a diagnosis of generalized MG made according to the following criteria: generalized subtype of myasthenia gravis with diffuse weakness, with or without ocular and respiratory involvement; decremental U-shaped response at 3-Hz repetitive nerve stimulation and/or increased jitter at single-fiber electromyography; exclusion of any other neurologic or inflammatory condition [20]. All patients were screened for the presence of thymoma by means of computed tomography or magnetic resonance imaging scanning of the mediastinum. The introduction or modification of immunomodulatory regimens (steroids, immunosuppressants, monoclonal antibodies, intravenous immunoglobulin (IVIG) or plasma exchange) and thymectomy in the last six months were further exclusion criteria. Prednisone was the main treatment (all patients), administered alone or in combination with cyclosporine (three patients, 21%), azathioprine (two patients, 14%), or mycophenolate mofetil (two patients, 14%). Fourteen patients were treated with prednisone since diagnosis for a median duration of 9.5 years (interquartile range (IQR) 14.3); two patients with a disease duration of 21 years had received cyclosporine for 19 years; three patients with a median disease duration of 4 years (IQR 15) had received azathioprine for 3 years (IQR 15); two patients with a disease duration of 12 and 18 years had received mycophenolate mofetil for 11 and 17 years, respectively. Finally, cholinesterase inhibitors had been employed in 11 patients (79%) since the diagnosis.

The disease severity, assessed using the modified Osserman classification [21], was established by two different neurologists who were both blinded to the plasmatic levels of MMPs and TIMPs. Five patients (36%) were classified as Osserman grade IIa, three patients (21%) as grade IIb, three patients (21%) as grade IIIa, two patients (14%) as grade IIIb, and only one patient (7%) as grade IV.

The blood samples from 13 age- and sex-matched healthy volunteers were collected among employees of the ‘Policlinico Paolo Giaccone’ University Hospital.

The study was performed according to the declaration of Helsinki and its later amendments and approved by the ‘Palermo I’ Ethical Committee. All patients included in the study consented to the use of their data for research purposes and signed informant consent for study participation.

### 2.2. Laboratory Testing

Plasma concentrations of gelatinases MMP-9 and MMP-2 and their inhibitors TIMP-1 and TIMP-2 were determined using the Human MMP-2 and MMP-9 enzyme-linked immunosorbent assay (ELISA) kit (Boster Biological Technology, Ltd., Pleasanton, CA, USA) and the Human TIMP-1 and TIMP-2 ELISA kit (Boster Biological Technology, Ltd., Pleasanton, CA, USA). Testing for autoantibodies against AChR was carried out by radioimmunoassay method (RIA) using a radio-receptor assay kit. Autoantibodies against MuSK were searched by ELISA in all AChR-negative patients.

### 2.3. Statistical Analysis

We reported continuous variables as median with IQR and categorical variables as numbers and related percentages. We compared categorical variables among groups using the Chi-square test and continuous variables by using the Mann–Whitney test. The Spearman correlation coefficient was calculated to evaluate any correlations among variables (plasmatic MMP-9, MMP-2, TIMP-1, TIMP-2, demographic and clinical data). We performed all tests using SPSS (v26) and established the level of significance as *p* < 0.05.

## 3. Results

### 3.1. Patients’ Population

We enrolled 14 patients affected by generalized myasthenia gravis (median 59 years, IQR 19; eight males, 57%) and 13 control subjects (median 48 years, IQR 9.5; five males, 38%). There was no significant difference between patients and controls for age (*p* = 0.09) and gender (*p* = 0.62). All included patients had a generalized subtype of MG with weakness of both ocular and non-ocular muscles with a median disease duration of 9.5 years (IQR 14.3). No patients with purely ocular symptoms were recruited. Twelve out of fourteen MG patients (86%) tested positive for anti-AChR antibodies in serum (median title 18 nmol/L, IQR 43.9), while no patients with anti-MuSK antibodies were enrolled. Hence, two patients resulted seronegative for both anti-AChR and anti-MuSK antibodies. An associated thymoma was found in five MG patients. All these patients underwent thymectomy within six months from the disease onset. Patients with and without thymoma did not differ with respect to age (*p* = 0.29), AChR antibodies serum levels (*p* = 0.18), disease duration (*p* = 0.89), gender (*p* = 0.36), as well as with regard to plasma levels of MMP-9 (*p* = 0.23), MMP-2 (*p* = 0.79), TIMP-1 (*p* = 0.69) and TIMP-2 (*p* = 0.79). All patients with thymoma had an AChR-positive form of generalized MG. There was no significant difference in the plasma levels of MMPs and TIMPs between patients with and without thymoma.

### 3.2. MMPs and TIMPs Plasma Levels

Table 1 and Figure 1 report plasma levels of MMPs and TIMPs along with MMP/TIMP ratios in patients and controls. The MMP-9 concentrations were much higher in the MG group compared to controls. Conversely, the MMP-2 levels were significantly reduced in the patients’ group. TIMP-1 concentrations were significantly lower in the MG group compared to controls, while no differences were found as regards the TIMP-2 concentrations. Finally, the MMP-9/TIMP-1 ratio was higher, while the MMP-2/TIMP-2 ratio was reduced in patients compared to controls.

Among MG patients, a significant direct correlation was demonstrated between MMP-2 levels and the severity of the disease (r = 0.65, *p* = 0.011). Instead, we showed only a correlation trend between MMP-9 levels and the disease severity (r = 0.49; *p* = 0.073). We also observed positive correlations between MMP-9 and TIMP-2 levels (r = 0.72, *p* = 0.003) and between MMP-2 levels and MMP-2/TIMP-2 ratio (r = 0.99, *p* = 0.0001). Finally, a negative correlation emerged between TIMP-1 levels and MMP-9/TIMP-1 ratio (r = −0.701, *p* = 0.005). Among controls, no significant direct correlations have been demonstrated among MMPs and TIMPs levels.

## 4. Discussion

In this study, we showed that the MMPs/TIMPs pattern was disrupted in patients with generalized MG compared to healthy controls. In particular, we observed a relevant increase in the plasma concentrations of MMP-9, along with a reduction in its inhibitor TIMP-1 in the patients’ group. Conversely, we detected a decrease in MMP-2 levels in MG patients compared to healthy subjects.

Elevated MMP-9 levels have been reported in several autoimmune diseases, such as systemic lupus erythematosus and multiple sclerosis [14,22], and mainly linked to the increased accumulation of blood cells [10]. Indeed, granulocytes and macrophages are considered to be major producers of MMP-9, that is, supposed to facilitate T-cell migration [3,14]. On these bases, it is thought that infiltrations of CD8+ T-lymphocytes, which are involved in the pathogenesis of MG, could be favored by an increased MMP-9 activity.

A study by Helgeland et al. described increased MMP-2 and MMP-9 levels in a patients’ cohort with generalized MG, thus supporting the link between the levels of MMPs and the pathogenesis of MG [19]. Our results partially differ from those reported by these authors, as we did not confirm the increase in MMP-2 levels, showing instead opposite findings of reduced MMP-2 levels in generalized MG. Different factors could explain such a discrepancy: (1) differently from our study, patients with generalized weakness as the presenting symptom were not included in the Helgeland’s work, where 27% of patients had a purely ocular form; (2) the authors did not included sex- and age-matched controls; (3) different therapeutic approaches could have influenced the expression of MMPs in our patients’ population.

In the present study, we evaluated plasma levels of MMPs and TIMPs in generalized MG patients, hypothesizing that they might be expression of disease activity. Indeed, a significant correlation was demonstrated only between MMP-2 levels and the severity of the disease, while there was only a trend of correlation between MMP-9 levels and the duration of the disease.

A previous study exploring the role of MMPs in several neuromuscular conditions before and after IVIG treatment showed that baseline elevated levels of MMP-9 dropped after IVIG, while a post-treatment increase in MMP-2 levels could indicate a relapsing disease [4]. Thus, while a reduction in MMP-9 concentrations after IVIG treatment could suggest tissue repair, an increasing in MMP-2 levels after IVIG could be related to relapsing tissue damage [4]. The present finding of reduced MMP-2 levels in MG compared to controls and the contemporary evidence of increased MMP-2 values in MG (i.e., with a tendency towards normalization) need to be explained. Based on our results, we can speculate that a MMP2 down-regulation could represent a compensatory beneficial mechanism towards the autoimmune pathogenetic mechanisms underlying the disease.

The negative correlation between TIMP-1 and MMP9/TIMP-1 ratio, together with the positive correlation between MMP-2 and MMP-2/TIMP-2 ratio suggests an impaired balance between MMPs and their inhibitors in generalized MG patients [2,4]. Instead, the positive correlation between MMP-9 and TIMP-2 could be in line with the previous hypothesis that a down-regulation of MMP-2 might compensate the autoimmune process underlying the disease. The lack of correlation between the presence of thymoma and MMP/TIMPs levels might be due to the limited number of patients enrolled, thus the link between MMPs/TIMPs pattern disruption and thymoma in MG need to be explored in future studies.

Several considerations and limitations of the present study should be taken into account. The principal limitation of the present study refers to the small sample size and the narrow phenotypic and serological spectrum of the patients’ population. Indeed, our group of patients was not representative of all MG patients, as we did not enroll neither patients with anti-MuSK antibodies nor patients with ocular subtype of MG. Additionally, the enrolled patients had a prolonged duration of disease. Thus, targeted studies on newly diagnosed generalized MG patients are warranted to clarify to what extent differences in the autoantibody profiles or in the clinical picture could influence the MMPs/TIMPs pattern.

Another consideration to be made is that in this study the levels of MMPs and TIMPs were assessed only at one time point, and functional evaluations of MMPs activity were not performed. Indeed, the plasma levels of MMPs could not strictly reflect the activity level of these enzymes, as active MMPs may be physiologically inactivated by different endogenous inhibitors, such as serum alpha-2 macroglobulin and alpha-2 antitrypsin [23]. Moreover, we are not able to rule out that changes in the plasma levels of MMPs and TIMPs could represent a consequence of inflammation rather than the direct expression of pathogenetic mechanisms. In this study, we did not measure MMP-3 concentrations, which may also have a role in generalized subtypes of MG as shown in a previous study [19]. The monitoring of MMPs and TIMPs levels over time and in different phases of the disease could offer new interesting clues to the pathogenesis of the disease and might provide useful tools to predict disease exacerbations or the response to different treatment strategies.

A last consideration has to be made on the possible influence of the immunomodulatory/immunosoppressive regimens on the plasma levels of MMPs and TIMPs. In particular, all patients had been treated with prednisone for an average of ten years. The effects of corticosteroids on MMPs levels are poorly understood. However, based on some experimental evidence, we may rule out that our findings were due to the steroid treatment [24,25]. Indeed, oppositely to our results, there is evidence that steroids can decrease MMP-9 and increase TIMP-1 levels [24,25], while MMP-2 and TIMP-2 levels may be not affected [26].

## 5. Conclusions

Our preliminary results may support the hypothesis that plasma levels of metalloproteinases and their inhibitors might play a role in the pathogenesis of generalized MG. However, future studies should be performed in wider groups of patients before and after treatment, both to confirm our findings and to assess the potential usefulness of MMPs and TIMPs evaluations as indicators of treatment response. Finally, it is noteworthy that synthetic inhibitors of MMPs have recently been produced and tested in several diseases. Thus, in the future, therapeutic approaches based on the use of inhibitors of MMPs in generalized MG could provide interesting clues to the pathogenetic role of an abnormal MMPs/TIMPs profile in MG.

## Figures and Tables

**Figure 1 brainsci-12-01439-f001:**
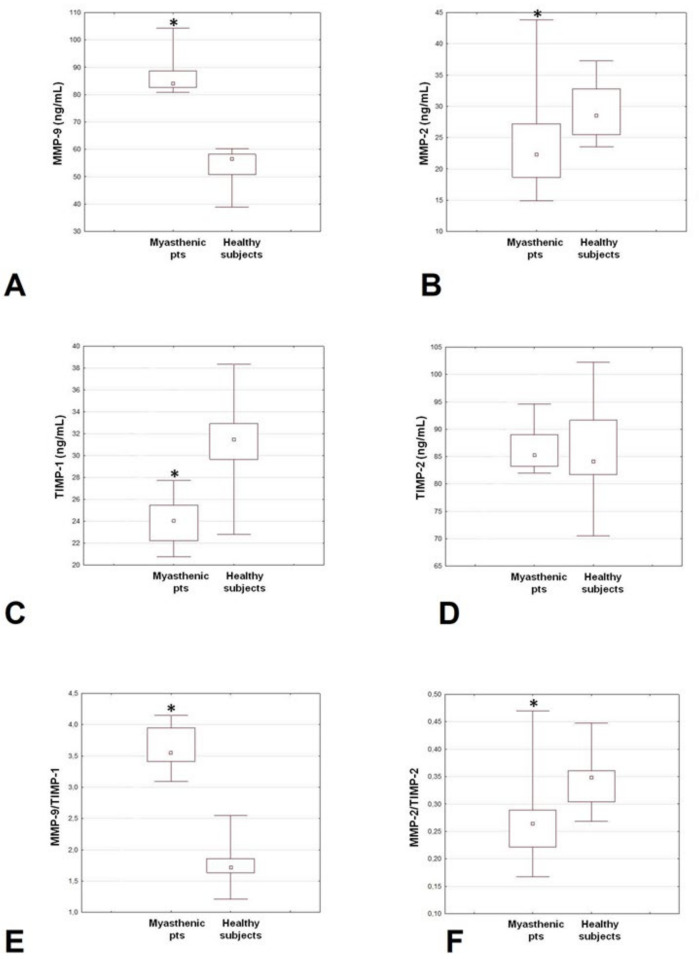
Plasma levels of MMP-9 (**A**), MMP-2 (**B**), TIMP-1 (**C**) and TIMP-2 (**D**), and MMP-9/TIMP-1 (**E**) and MMP-2/TIMP-2 (**F**) ratios in patients with generalized MG patients and controls. MG: myasthenia gravis; MMP: matrix metalloproteinases; TIMP: tissue inhibitors of metalloproteinases. * *p* < 0.05.

**Table 1 brainsci-12-01439-t001:** Clinical data and MMP and TIMP plasmatic levels in MG patients and controls.

	*MG Patients (n = 14)*	*Controls (n = 13)*	*p Value*
Age (years)	59 (19)	48 (9.5)	0.0941
Male, *n* (%)	8 (57%)	5 (38%)	0.62
Anti-AChR Ab+	12 (86%)	/	/
AChR antibodies titres (nmol/L)	18 (43.9)	/	/
Disease duration (years)	9.5 (14.3)	/	/

AChR Ab, acetylcholine receptor antibodies; MMP, matrix metalloproteinases; MG, myasthenia gravis; TIMPs, tissue inhibitors of metalloproteinases. Continuous variables are expressed as median (interquartile ranges), categorical variables are expressed as percentages.

## Data Availability

Data are available from the corresponding author upon reasonable request.

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
