# Peer review of "Metalloproteinases and Tissue Inhibitors in Generalized Myasthenia Gravis. A Preliminary Study"

_brainsci, 2022, doi:10.3390/brainsci12111439_

Round 1

Reviewer 1 Report

The authors analyzed the levels of plasmatic MMP-2 and its inhibitor TIMP-2, as well as MMP-9 and its inhibitor TIMP-1 in patients with myathenia gravis and health controls, aiming at identifying new biomarkers for the disease.

The study was approved by a local ethics committee. Appropriate consent was obtained from patients.

One point, however, need clarification:

1. Why was the control group selected among employees of the ‘Policlinico Paolo Giaccone’ University Hospital? Wouldn't the employment relationship make them feel obligated to participate? Why not select the control group from the general population?

Please consider the following suggestions:

2. indicate in section 2.1 the number of patients included in the study (include absolute numbers, not only the %, because of the small sample size);

3. do not present the same results in the table 1 and figure 1, keep clinical data in the table and MMP and TIMP plasmatic leves in the figure;

4. include in the results the correlation analyses of the plasmatic measurements obtained from the control group;

5. include information regarding patients with thymoma in the results section, the text “All patients with thymoma had an AChR-positive form of generalized MG. There was no significant difference between the plasma levels of MMPs and TIMPs and the presence of thymoma.” is in line 198 in the discussion section;

Please also review the following texts:

6. lines 23 to 25 (Abstract) the text “An increase in MMP-9 levels and MMP-9/TIMP-1 ratio, along with a decrease in MMP-2 levels and MMP-2/TIMP-2 ratio were detected in patients with generalized MG. Higher levels of MMP-2 correlated with greater disease severity” describes results, not discussion;

7. line 183 – please review grammar “... we evaluated plasma levels of MMPs and TIMPs in generalized 182 MG patients in the hypothesis that they might be expression of disease activity.”

8. line 193 – please review grammar “... evidence that more increased MMP-2 values in MG...”.

Reviewer 2 Report

The authors conducted a preliminary study to evaluated the plasma levels of MMP-9 and MMP-2 and their tissue inhibitors TIMP 1 and TIMP-2 in a cohort of generalized MG. They found an increase in MMP-9 levels and MMP-9/TIMP-1 ratio, along with a decrease in MMP-2 levels and MMP-2/TIMP-2 ratio in patients with generalized MG. Higher levels of MMP-2 correlated with greater disease severity. They concluded that their findings suggest that MMPs and TIMPs could play a role in the pathogenesis of MG and might be associated with the risk of clinical deterioration. I have some major concerns.

1.      The study enrolled 14 patients affected by generalized MG and 13 control subjects. All included patients had a median disease duration of 9.5 years (IQR 14.3). Since the disease duration of the patients were long, I would doubt if the plasma levels of MMP and TIMP in patients with newly diagnosed generalized MG will be the same as the results of the studies. Did the authors also enroll patients with newly MG?

2.      The study only test the plasma levels of MMP and TIMP of patients with MG for one time during their disease duration. It is very important, if possible, to evaluate the change of the plasma level of MMP and TIMP of patients with MG during disease duration.

3.      As the authors discussed, the study had two major limitations. One was the small sample size. The other was that the patients had been treated with steroids before testing of MMP and TIMP. Therefore, it is impossible to determine whether the changes in the plasma levels of MMPs and TIMPs could represent a consequence of inflammation rather than the direct expression of pathogenetic mechanisms. They should be cautious to draw the conclusion. 

Round 2

Reviewer 2 Report

The authors have addressed all my concerns.